# An Analytical Model for Evaluation of the Properties of Metallic Coatings in RF Structures

**Giovanni Castorina [1,2,\*], Augusto Marcelli [1,3], Francesca Monforte [4], Stefano Sarti [5] and Bruno Spataro [1]**

1    INFN—Laboratori Nazionali di Frascati, Via E. Fermi 40, 00044 Frascati, Italy;
     augusto.marcelli@lnf.infn.it (A.M.); bruno.spataro@lnf.infn.it (B.S.)
2    Dipartimento di Ingegneria Elettrica, Elettronica e Informatica, University of Catania, 95126 Catania, Italy
3    RICMASS, Rome International Center for Materials Science Superstripes, Via dei Sabelli 119A,
     00185 Rome, Italy
4    CNR Istituto per la Microelettronica e Microsistemi—IMM, VIII Strada 5, 95121 Catania, Italy;
     marzia.monforte@gmail.com
5    Dipartimento di Fisica, University of Rome Sapienza, 00185 Rome, Italy; stefano.sarti@roma1.infn.it
*    Correspondence: giovanni.castorina@lnf.infn.it; Tel.: +39-6-9403-2253

**Abstract:** A simple analytic model based on the equations of the propagation matrices theory has been developed in order to evaluate the effective skin depth of coated metallic surfaces. With particular attention to the R&D of highly-performing accelerating structures, different thick coatings with excellent mechanical and electrical properties have been considered such as molybdenum and its oxides, p-doped SiC, and TiN. Calculations show that copper coated with a p-type SiC may exhibit an improved hardness and a higher thermal resistance. Combined with experimental tests, this study may support the identification of reliable multilayers capable of improving the higher power performance of radio-frequency (RF) structures in terms of the accelerating gradient in order to increase the resistance to the high thermal stress of structures made with copper.

**Keywords:** analytical model; transition metal (TM) coating; accelerating structures; silicon carbide

## 1. Introduction

Coatings are used in several technological and industrial applications. Metallic coatings, in particular, are used in aerospace and space applications to increase the lifetime of critical components or to enhance the resistance to friction, abrasion, corrosion, etc. In space, in high power radio-frequency (RF) devices, surface properties govern the secondary electron emission (SEE) from materials and are at the origin of the resonant avalanche electron discharge known as the multipactor effect [1–4]. The latter limits the attainable power of these devices. The understanding of the properties of metallic surfaces and coatings characterized by a low secondary electron emission yield (SEY) for steel (large accelerators) and aluminium for space applications is another important field of interest.

Their properties strongly depend on the substrate, the material, its microstructure, and process parameters such as temperature, atmosphere, and time of annealing. Typical substrates are semiconductors like silicon or insulating crystals such as quartz with different orientation. However, high-conductivity metallic coatings on metallic substrates are also an interesting option for many applications, and in particular, for high performance accelerator components where one of the main issues is the RF breakdown [5]. Practically, experimental tests can be carried out comparing the response of different materials with different thickness and process parameters to identify the optimal thickness and the dependency by growing parameters and post-deposition annealing procedures. In this

framework, in recent years modelling and simulation tools have become more and more important, being effective alternatives to long and costly experimental tests [6,7]. This contribution presents and discusses a simple analytical model suitable for the simulation of the properties of thick metallic coatings, such as those considered to improve/optimize performances of accelerator components.

## 2. Analytical Model

In this section, we describe the model developed for the evaluation of the effective skin depth of a metallic substrate coated with a relatively thick conducting layer. This simple tool could be used in order to estimate the reduction of the quality factor of an accelerating structure coated with a metallic layer. The model uses the well-known equations of the propagation matrices theory [8] applied to a system composed of an infinite metallic bulk with no reflection and a dielectric or metallic coating. The electromagnetic fields and the effective skin depth in this bulk–coating multilayer structure have been evaluated in order to estimate the quality factor and the surface strength of an ideal coated device, such as an RF cavity. Oxygen-free electronic copper (OFE copper) is the material most commonly used in normal conducting accelerator structures (C10100 in the copper unified numbering system). However, to increase the accelerating gradient of RF cavities working at higher frequencies, we may consider different manufacturing techniques (e.g., electroforming or electron beam welding), and the use of alternative materials such as molybdenum sputtered on copper, different copper alloys, and the realization of single and multi-layer surfaces with precision-controlled properties [9].

To consider RF applications at high frequency, the analytical model has been also extended up to the THz range, adopting the Drude model of transition metal (TM) oxides for the coating region [10]. In this model, the bulk material is considered infinite, so no field is reflected, and the metallic layer has a thickness equal to $\Delta x$. The external vacuum is assumed as the first medium.

The position of the bulk–coating interface shown in Figure 1 is $x_{int}$, while that of the vacuum-coating is set to 0. Because not all quantities are continuous at the interface, it is also useful to show the positions of $0^{\pm}$ and $x_{int}^{\pm}$ at the interfaces. The propagation constants $\gamma_{0,c,b}$ and the medium impedance $\eta_{0,c,b}$ for vacuum, coating, and bulk are described by Equations (1)–(6), respectively. The good conductor approximation formulas are used for the bulk. No assumptions are made for the coating.

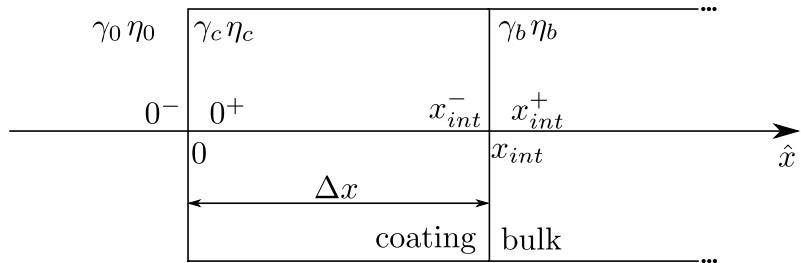

**Figure 1.** Schematic view of the model for a layer coated on top of a metallic substrate (on the right side of the figure).

$$\gamma_0 = \omega\sqrt{\epsilon_0\mu_0} \tag{1}$$

$$\gamma_c = \alpha + j\beta = \omega\sqrt{\frac{\mu\epsilon_0\epsilon_{rc}}{2}\left[\sqrt{1 + \frac{\sigma_c^2}{(\epsilon_0\epsilon_{rc}\omega)^2}} + 1\right]} + j\omega\sqrt{\frac{\mu\epsilon_0\epsilon_{rc}}{2}\left[\sqrt{1 + \frac{\sigma_c^2}{(\epsilon_0\epsilon_{rc}\omega)^2}} - 1\right]} \tag{2}$$

$$\gamma_b = (1+j)\sqrt{\frac{\omega\mu_0\mu_{rb}\sigma_b}{2}} \tag{3}$$

$$\eta_0 = \sqrt{\frac{\mu_0}{\epsilon_0}} \tag{4}$$

$$\eta_c = \eta_0 \sqrt{\frac{j\omega\epsilon_0 \mu_{rc}}{\sigma_c + j\omega\epsilon_0 \epsilon_{rc}}} \tag{5}$$

$$\eta_b = (1+j)\eta_0 \sqrt{\frac{\omega\epsilon_0 \mu_{rb}}{2\sigma_b}} \tag{6}$$

Assuming no reflection from the bulk, $\Gamma(x_{int}^+) = 0$, so that the wave impedance (Equation (7)) in the bulk is just the medium impedance:

$$Z(x_{int}^+) = \frac{E(x_{int}^+)}{H(x_{int}^+)} = \eta_b \frac{1 + \Gamma(x_{int}^+)}{1 - \Gamma(x_{int}^+)} = \eta_b \tag{7}$$

the latter is a continuous function at the interface:

$$Z(x_{int}^-) = Z(x_{int}^+) \tag{8}$$

while the reflection coefficient is not a continuous function:

$$\Gamma(x_{int}^-) = \frac{Z(x_{int}^-) - \eta_c}{Z(x_{int}^-) + \eta_c} \tag{9}$$

The reflection coefficient propagates at $0^+$ with an exponential behavior:

$$\Gamma(0^+) = \Gamma(x_{int}^-)\, e^{-2\eta_c \Delta x} \tag{10}$$

and due to the continuity of the wave impedance function, $Z(0^-) = Z(0^+)$, and the wave impedance at $0^+$ can be evaluated as in Equation (7):

$$Z(0^+) = \eta_c \frac{1 + \Gamma(0^+)}{1 - \Gamma(0^+)} \tag{11}$$

while the reflection coefficient at the vacuum–coating interface is

$$\Gamma(0^-) = \frac{Z(0^-) - \eta_0}{Z(0^-) + \eta_0} \tag{12}$$

and if the incident field, $E_{inc}(0^-)$, is normalized to 1 (V/m). The reflected electric field can be evaluated as:

$$E_r(0^-) = \Gamma(0^-)\, E_{inc}(0^-) \tag{13}$$

the total field in vacuum is then the sum of the incident and the reflected field:

$$E(0^-) = E_{inc}(0^-) + E_r(0^-) \tag{14}$$

and since the electric field is continuous at the interface:

$$E(0^+) = E(0^-) \tag{15}$$

Finally, the electric field in the bulk and in the coating layer can be calculated using Equations (16) and (17):

$$E_c(x) = E(0^+)\, e^{-\Re(\gamma_c)x} \quad \text{for} \ \ 0 < x \leq x_{int} \tag{16}$$

$$E_b(x) = E(x_{int})\, e^{-\Re(\gamma_b)x} \quad \text{for} \ \ x > x_{int} \tag{17}$$

where $\Re(\gamma_c)$ is the real part of $\gamma_c$.

## 3. Results and Discussion

The model allows effective skin depth ($\delta_{eff}$) to be evaluated, which is the distance where the field is $1/e$ of the value at $0^+$ point. The quality factor $Q = \omega$ (Stored Energy/Power Dissipation) is inversely proportional to the skin depth, so a small value for $\delta_{eff}$ is desirable in order to get a resonant cavity with a high Q. Furthermore, if the coating is a good conductor, the electric field on the coating–copper interface can be reduced by a factor two or more, depending on the thickness and electrical properties of the protective layer. In this way, an accelerating RF structure with such a coating should exhibit improved characteristics, such as a longer lifetime.

Combining Equations (16) and (17), it is easy to show that unless $\Re(\gamma_c)x_{int} \geq 1$, the effective penetration depth (defined as $E(x = \delta_{eff}) = E(0)/e$) is defined by the relation:

$$(\Re(\gamma_c) - \Re(\gamma_b))x_{int} + \Re(\gamma_b)\delta_{eff} = 1 \Rightarrow \delta_{eff} = x_{int} + \frac{1 - \Re(\gamma_c)x_{int}}{\Re(\gamma_b)} \tag{18}$$

(if $\Re(\gamma_c)x_{int} \geq 1$, one has $\delta_{eff} \leq x_{int} = 1/\Re(\gamma_c)$). Thus, the larger $\Re(\gamma_c)$, the lower will be $\delta_{eff}$. On the other hand, Equation (2) reduces to Equation (1) when $(\sigma_c/\epsilon_0\epsilon_{rc}\omega) \gg 1$, which is always satisfied at microwave frequencies unless $\sigma$ is very small ($\leq 10^2$ $S/m$) and $\epsilon_{rc}$ is very large ($\geq 10$). This is never the case for the materials considered in this study, where conductivity is greater then $10^4$ ($S/m$) and relative permeability below 10. Thus, $\Re(\gamma_c) \simeq \sqrt{\mu_0\omega\sigma_c}$, and high values of $\Re(\gamma_c)$ corresponds to high values of $\sigma_c$.

For a low conductivity coating, the reflected wave is significantly reduced, and then more transmitted power has to be dissipated in the coating region. If $\sigma_c$ (or the coating thickness) is not sufficiently large, the electric field at the interface with copper can be higher than a copper surface without coating (Figure 2). For the trivial case with a $\sigma_c$ greater than the conductivity of copper inside, in the bulk, the electric field is always lower. However, it is not an easy task to identify an alloy with a high conductivity combined with improved mechanical, thermal, and chemical properties. For the common case of $\sigma_c$ lower than copper conductivity, it is possible to use the analytical tool proposed here to calculate the minimum thickness required to obtain an electric field on the copper surface equal to or lower than the field that would be present on copper without any coating. This value can be extremely large and is not always feasible. In this case, a trade-off between thermal and electrical performance must occur.

An optimized coating could be extremely useful to mechanically and chemically stabilize the copper bulk (see below). From thermal and hardness points of view, significant advantages could be achieved if a large fraction of the heat would be dissipated in the coating layer instead of in the bulk. Within the proposed model, it is straightforward to calculate the energy dissipated in both coating and bulk using the following equations:

$$W_c = \int_0^{x_{int}} \sigma_c E^2 dx = E(0)^2 \frac{\sigma_c}{\Re(\gamma_c)} [1 - \exp(-2\Re(\gamma_c)x_{int})] \tag{19}$$

$$W_b = \int_{x_{int}}^{\infty} \sigma_b E^2 dx = E(0)^2 \frac{\sigma_b}{2\Re(\gamma_b)} \exp(-2 \times \Re(\gamma_c)x_{int}) \tag{20}$$

whose ratio is

$$\frac{W_c}{W_b} = \frac{\sigma_c \Re(\gamma_b)}{\sigma_b \Re(\gamma_c)} \frac{1 - \exp(-2\Re(\gamma_c)x_{int})}{\exp(-2 \times \Re(\gamma_c)x_{int})} \simeq \sqrt{\frac{\sigma_c}{\sigma_b}} [\exp(2\sqrt{\mu_0\omega\sigma_c}x_{int}) - 1] \tag{21}$$

where in the last step, $\Re(\gamma_c) \simeq \sqrt{\mu_0\omega\sigma_c}$ has been assumed, as previously discussed. Again, the larger the coating conductivity $\sigma_c$, the higher the fraction of energy (and the associated heat) dissipated in the coating (see Figure 2). Due to the exponential term, a small increase in the conductivity $\sigma_c$ may lead to a significant increase of the heat trapped in the coating, and thus a considerable decrease of the fraction of heat dissipated in the copper bulk. An ideal material for the coating is thus a system

with a large thermal resistance and a sufficiently high conductivity to ensure a large fraction of heat generation in the coating layer.

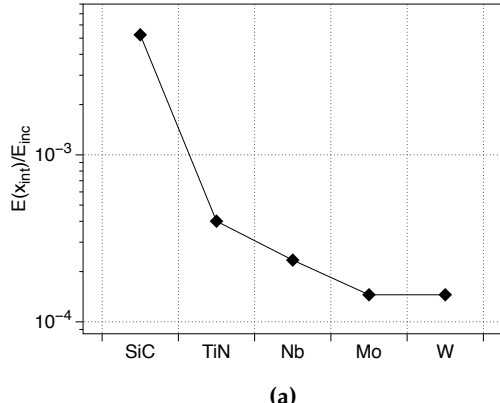
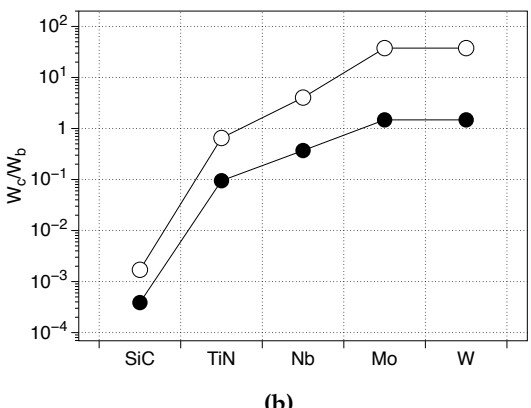

(a)                                    (b)

**Figure 2.** (**a**) Comparison of the ratio of the electric field at the coating–bulk interface and the incident field. (**b**) The dissipated power in both coating layer and bulk for different materials: calculations refer to 11 GHz (filled circles) and to 0.2 THz (empty circles) and to 500 nm coating thickness.

At higher frequencies, the approximation $\Re(\gamma_c) \simeq \sqrt{\mu_0 \omega \sigma_c}$ may fail. In this case, the value of $\epsilon_{rc}$ may play a role in the determination of $\Re(\gamma_c)$. To maximize $\Re(\gamma_c)$, we may change either $\sigma_c$, $\epsilon_{rc}$, or both. However, for all coatings, we considered the contribution to $\epsilon_{rc}$ to be negligible (<1%), even at the highest frequency considered (200 GHz). Thus, the above considerations are valid for all coatings and frequencies.

## 4. Materials and Methods of Deposition

The choice of coating material for accelerator components such as RF cavities depends on several factors, such as conductivity, mechanical resistance, and chemical affinity to the bulk. Compared to copper, an ideal coating material has to show comparable or superior mechanical and chemical-physical properties in order to tolerate the thermal stress (pulse heating) induced by multi-megawatt high frequency electromagnetic fields, a high/moderate electrical conductivity, and a superior chemical inertness. Furthermore, the deposition process of a coating layer should be capable of growing homogeneous non-porous thick layers on complex curved surfaces and to guarantee a conformal coverage, compatible to copper bulk. In addition to Nb and Mo layers which exhibit excellent properties, other promising materials are metallic silicon carbide (6H-SiC) and titanium nitride (TiN).

SiC is a semiconductor, and although the metallic conductivity can be enhanced by heavy doping with B or Al [11], the maximum value obtainable is roughly $10^4$ S/m, a value associated with poor electrical characteristics. However, this material is interesting due to its high melting point (2730 °C), low thermal expansion, hardness (9 on mohs scale), and chemical affinity with copper. Among the possible deposition methods, the Inductively Coupled Plasma Chemical Vapor Deposition (ICP CVD) allows the synthesis of SiC layers over larger areas with very high thickness homogeneity and roughness below 10 nm [12]. It is a non-thermal synthesis suitable for not-brazed hard copper structures (temperature below 100 °C), and the vertical geometry of the system guarantees the presence of a highly-ionized plasma. TiN is a refractory material (2930 °C melting point) with much higher electrical conductivity ($2 \times 10^6$ S/m) than SiC. It is an extremely hard ceramic material with good mechanical properties. Thin titanium nitride films can be obtained through sputter deposition. Thanks to this technique, it is possible to deposit homogeneous samples over large areas and to cover the internal surface of complex devices with thicknesses from hundreds of nm to a few μm.

The analytical model above described has been used to evaluate the effective skin depth of different possible coating materials. In Figure 3, the electric field profile for different coatings is shown. The electrical properties of the doped SiC are roughly the same as the MoO$_3$—both are insulators. The electrical conductivity of TiN is one order of magnitude lower than a good conductor; thus, looking at Figure 3, the skin depth value is in between doped SiC and Mo. The best value for $\delta_{eff}$ is obtained with TM metals like Mo. However, the effective skin depth of MoO$_3$ and doped SiC is only 30% greater then Mo; i.e., the quality factor is 30% lower—a reduction value tolerable for common accelerating structures.

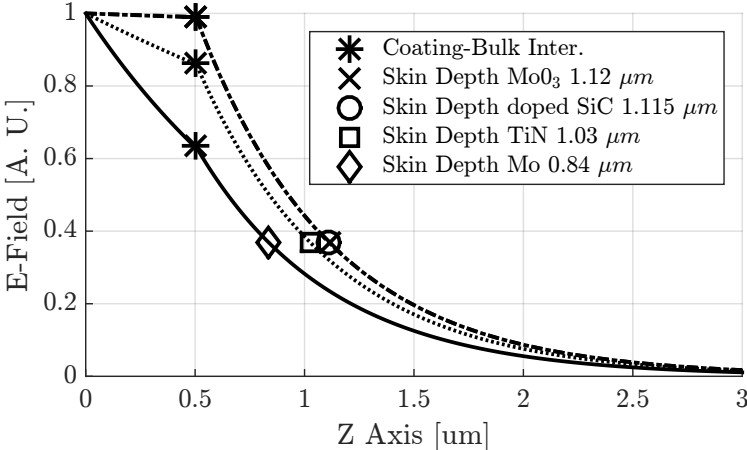

**Figure 3.** Comparison of the electric field profiles of a copper bulk coated with a 500 nm-thick layer of MoO$_3$ (**- - -**), SiC (**-·-·**), TiN (**····**), and Mo (**——**).

A possible alternative to these single materials is represented by the growth of multilayers obtained by combining a metallic material (such as W and Mo [13]) with a thin semiconductor layer (e.g., SiC) in order to improve thermal strength and mechanical properties.

## 5. Conclusions

An analytical model based on transmission line equations has been developed in order to evaluate the effective skin depth of thick coatings on a metallic substrate such as copper. This tool allows a first evaluation of the quality factor of an RF device, of the electric field, and of the dissipated power in the coating layer.

Different materials for the coating of a copper surface have been considered, such as SiC, TiN, Mo, and its oxides. The best electrical performance can be achieved with TM metals like Mo. P-doped SiC and MoO$_3$ are also suitable for their mechanical properties, with a tolerable reduction of the quality factor. As an example, a conducting layer of SiC could be deposited on copper through a ICP-CVD process—a low temperature process which is compatible with the manufacture of hard copper non-brazed structures. However, a good trade off between electrical and mechanical performance can be also obtained with TiN.

The analytical model we presented is a first step in this approach that combines experimental tests and modelling. It will be improved by taking a multilayer setup into account, in particular to include intermediate materials for the enhancement of adhesion between the bulk and the coating layer. An important limitation is that it considers ideal surfaces without transition regions, which should be taken into account when very thin layers are considered. At present, other important surface effects such as roughness and slope errors are also not considered, and have to be included in the model. In particular, since the Q of an RF cavity depends on the electrical resistivity of the surface,

and roughness and cracks affect the RF properties of a device, the behaviour of a real surface can be significantly different from that of a highly smooth surface.

**Acknowledgments:** Giovanni Castorina gratefully acknowledges INFN for the two years fellowship.

**Author Contributions:** Augusto Marcelli and Giovanni Castorina conceived the research; Giovanni Castorina and Stefano Sarti wrote the code and Giovanni Castorina, Augusto Marcelli, Francesca Monforte, Stefano Sarti and Bruno Spataro wrote the paper. All authors have read and approved the final manuscript.

**Conflicts of Interest:** The authors declare no conflict of interest.

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
