# Peer review of "An Analytical Model for Evaluation of the Properties of Metallic Coatings in RF Structures"

_condensedmatter, doi:10.3390/condmat1010012_

Reviewer 1 Report

The article is well organized and presented. Recommend to publish it in Condensed Matter.

Author Response

We thank the first referee for his appreciation.

Reviewer 2 Report

The manuscript presents a model for metallic coating, based on electric field reflectance of a coating/substrate interface. The study purposes to achieve a thick coating for accelerating structures, in special RF cavity. The main points should be evaluated:

General comments:

·         A special attention for the application should be taken, for example, citing examples commercial materials and methods of coatings used in these components. Highlight the limitations of them, and how the study could contribute to overcome it.

·         Define clearly each variable presented in the equations, such as skin depth, quality factor and wave impedance.

·         Include more references in order to support the importance and results of the study.

·         Insert a result and discussion, acknowledgements and abbreviations section, following the Journal’s guidelines.

·    As suggestion, the title should be more specific, stating for what kind of application the analytical model is targeting. Once metallic coating requirements might be different in other applications (e.g. porous coating in biomedical field).

·         In the conclusion section, include details about some weakness of the analytic model and how to improve it (e.g. multilayer coating).

Specific comments:

·         Line 12: cite some general examples of application where metallic coatings are applied.

·         Line 13: add “atmosphere” as process parameters for coating.

·         Line 16: insert examples of high performance accelerator components, and what kind of coatings is required (e.g. thick metallic coatings)

·         Line 32: insert more details about what kind of materials RF cavities are made.

·         Line 59: highlight the reason to study copper as substrate.

·         Line 67: Support the sentence “… which is never the case for the materials considered in this study.” with numbers.

·         Line 70: the phrase “… it might seem that no coating solution is helpful.” Is confused. Please, rewrite the sentence.

·         Line 78: is the Fig. 2 a result or compilation of published data?

·         Line 90: For “several factors.” include more details.

·         Line 104: For “good roughness”, include more details.

·         Line 110: Insert more information about the use of Mo, Nb and W, in the same way that were written for TiN and SiC layers.

·         Line 119-121: State clearly the limitations of the study and include details about the multilayer model proposed.

Author Response

The manuscript presents a model for metallic coating, based on electric field reflectance of a coating/substrate interface. The study purposes to achieve a thick coating for accelerating structures, in special RF cavity. The main points should be evaluated:

General comments:

·     A special attention for the application should be taken, for example, citing examples commercial materials and methods of coatings used in these components. Highlight the limitations of them, and how the study could contribute to overcome it.

Response: Metallic coatings, in particular, are used in aerospace and space applications for increasing lifetime of critical components or to enhance the resistance to friction, abrasion, corrosion, etc. In space, in high power RF devices surface properties govern the secondary electron emission (SEE) from materials and are at the origin of the resonant avalanche electron discharge known as the multipactor effect. The latter limits the attainable power of these devices. The understanding of the properties of metallic surfaces and coatings characterized by a low secondary electron emission yield (SEY) for steel (large accelerators) and alluminium for space applications is another important field of interest. Few sentences have been added in the introduction.

·         Define clearly each variable presented in the equations, such as skin depth, quality factor and wave impedance.

Response: We improved the definition of all variables.

·         Include more references in order to support the importance and results of the study.

Response: Few references have been added in the text regarding the relevance of this topic in the RF breakdown.

·         Insert a result and discussion, acknowledgements and abbreviations section, following the Journal’s guidelines.

Response: We modified the text according to the Journal’s guidelines.

·     As suggestion, the title should be more specific, stating for what kind of application the analytical model is targeting. Once metallic coating requirements might be different in other applications (e.g. porous coating in biomedical field).

Response: We modified the title as ” An analytical model for evaluating the properties of metallic coatings in RF structures”

·         In the conclusion section, include details about some weakness of the analytic model and how to improve it (e.g. multilayer coating).

Response: We modified the conclusion section including some comments on the weakness of the analytic model.

Specific comments:

·         Line 12: cite some general examples of application where metallic coatings are applied.

Response: Metallic coatings, in particular, are used in aerospace and space applications for increasing lifetime of critical components or to enhance the resistance to friction, abrasion, corrosion, etc. In space, in high power RF devices surface properties govern the secondary electron emission (SEE) from materials and are at the origin of the resonant avalanche electron discharge known as the multipactor effect. The latter limits the attainable power of these devices. The understanding of the properties of metallic surfaces and coatings characterized by a low secondary electron emission yield (SEY) for steel (large accelerators) and alluminium for space applications is another important field of interest. Few sentences have been added in the introduction.

·         Line 13: add “atmosphere” as process parameters for coating.

Response: Added.

·         Line 16: insert examples of high performance accelerator components, and what kind of coatings is required (e.g. thick metallic coatings)

Response: Few sentences have been added to the introduction.

·         Line 32: insert more details about what kind of materials RF cavities are made.

Response: Oxygen free electronic copper (OFE copper) is the material most commonly used in normal conducting accelerator structures (C10100 in the copper unified numbering system.) However, to increase the accelerating gradient of RF cavities working at higher frequencies we may consider different manufacturing techniques, e.g., electroforming or electron beam welding, and the use of alternative materials such as molybdenum sputtered on copper, different copper alloys, the realization of single and multi-layer surfaces with precision-controlled properties, etc.

·         Line 59: highlight the reason to study copper as substrate.

Response: see the previous comment.

·         Line 67: Support the sentence “… which is never the case for the materials considered in this study.” with numbers.

Response: “….. where conductivity is greater then 104 S/m and relative permeability below 10 “ .

·         Line 70: the phrase “… it might seem that no coating solution is helpful.” Is confused. Please, rewrite the sentence.

Response: The sentence has been rewritten as: “since it is difficult to find a material with a conductivity higher than copper, and with comparable or better mechanical properties, a simple coating solution is not easy to identify”

·         Line 78: is the Fig. 2 a result or compilation of published data

Response: It is an original image never published before.

·         Line 90: For “several factors.” include more details.

Response: “ ... such as conductivity, mechanical resistance and chemical affinity to the bulk.”

·         Line 104: For “good roughness”, include more details.

Response: “...and a roughness<10 nm”

·         Line 110: Insert more information about the use of Mo, Nb and W, in the same way that were written for TiN and SiC layers.

Response: Few sentences have been added.

·         Line 119-121: State clearly the limitations of the study and include details about the multilayer model proposed.

Response: few sentences have been added to the introduction. 

Reviewer 3 Report

Concerning the novelty and significance, I am not sure this paper is eligible to the journal of Condensed Matter, and it can be at least improved based on the comments that I have as follows.

1.      Most fundamental work on which this paper based were established many years ago (see the Ref 3 and 4). What’s new here? It looks not much scientific significance in this paper.

2.      It’s ok not to introduce the well-known physical parameters likeεandμ to the reader, but the subscripts used are messy (b,c,rb,rc), i.e., they do not keep the consistency through the manuscript.

3.      R(γ) shown in equation (16) should be defined, otherwise, the reader who is not in this field will not understand it.

4.      Coating thickness/xint is missing in Fig2.

Author Response

Concerning the novelty and significance, I am not sure this paper is eligible to the journal of Condensed Matter, and it can be at least improved based on the comments that I have as follows.

1.        Most fundamental work on which this paper based were established many years ago (see the Ref 3 and 4). What’s new here? It looks not much scientific significance in this paper.

Response: We agree with the referee comment that the underlying physics it’s well known. However, also the technology of RF devices is still that of several decades ago. On the contrary, the demand is continuously increasing for industrial, medical and scientific accelerators. Since, at present there are no competitive alternatives to cavities made in (OFE) copper, the application of metallic coatings to RF cavity is a new and stimulating route to improve the existing technology. This analytical tool could be extremely useful to plan experimental tests and for preliminary comparison with experimental results.

2.      It’s ok not to introduce the well-known physical parameters likeεandμ to the reader, but the subscripts used are messy (b,c,rb,rc), i.e., they do not keep the consistency through the manuscript.

Response: Subscripts have been corrected through the manuscript. 

3.      R(γ) shown in equation (16) should be defined, otherwise, the reader who is not in this field will not understand it.

Response: Defined

4.      Coating thickness/xint is missing in Fig2.

Response: Added

Round  2

Reviewer 2 Report

The content of the manuscript was successfully improved. The title and introduction sections are clear and understandable, the symbols are well defined along the text, and the discussion and conclusion sections are concise and direct. The manuscript presents good quality for publication.

Reviewer 3 Report

I am satisfied with the revised version and suggest to publish it as it is now.